# Human Metapneumovirus: Etiological Agent of Severe Acute Respiratory Infections in Hospitalized and Deceased Patients with a Negative Diagnosis of Influenza

**DOI:** 10.3390/pathogens9020085

**Published:** 2020-01-28

**Authors:** Gisela Barrera-Badillo, Beatriz Olivares-Flores, Adriana Ruiz-López, Miguel Ángel Fierro-Valdez, Rosaura Idania Gutiérrez-Vargas, Irma López-Martínez

**Affiliations:** 1Institute of Epidemiological Diagnosis and Reference “Dr. Manuel Martinez Baez” (InDRE), Health Secretary. Francisco de P. Miranda No. 177. Colony Lomas de Plateros. Town Hall., Alvaro Obregon 1480, Mexico City, Mexico; beatriz.olivares@salud.gob.mx (B.O.-F.); adryrulams_w@hotmail.com (A.R.-L.); mick-angel_09@hotmail.com (M.Á.F.-V.); lopezmi74@gmail.com (I.L.-M.); 2General Directorate of Epidemiology; Health Secretary. Francisco de P. Miranda No. 157. Colony Lomas de Plateros. Town Hall. Alvaro Obregon, Mexico City 01480, Mexico; rosaurafluen@gmail.com

**Keywords:** human metapneumovirus, epidemiology, respiratory viruses

## Abstract

Human metapneumovirus (HMPV) is one of the four major viral pathogens associated with acute respiratory tract infections (ARI) and creates a substantial burden of disease, particularly in young children (<5 years) and older individuals (≥65 years). The objective of this study was to determine the epidemiological behavior of HMPV in Mexico. This retrospective study was conducted over a nine-year period and used 7283 influenza-negative respiratory samples from hospitalized and deceased patients who presented Severe Acute Respiratory Infection (SARI). The samples were processed with the help of qualitative multiplex RT-PCR for simultaneous detection of 14 respiratory viruses (xTAG^®^ RVP FAST v2). 40.8% of the samples were positive for respiratory viruses, mainly rhinovirus/enterovirus (47.6%), respiratory syncytial virus (15.9%), HMPV (11.1%) and parainfluenza virus (8.9%). Other respiratory viruses and co-infections accounted for 16.5%. HMPV infects all age groups, but the most affected group was infants between 29 days and 9 years of age (65.6%) and adults who are 40 years and older (25.7%). HMPV circulates every year from November to April, and the highest circulation was observed in late winter. The results of this study aim to raise awareness among clinicians about the high epidemiological impact of HMPV in young children and older individuals in order to reduce the economic burden in terms of health care costs.

## 1. Introduction

Acute respiratory infections (ARIs) pose a significant public health issue worldwide. There are more than 200 respiratory viruses that can cause ARIs. Influenza, respiratory syncytial virus, human rhinovirus and human metapneumovirus (HMPV) are the most common viral agents associated with ARIs, accounting for around 70% of all ARIs [1]. All these viral pathogens are associated with acute lower respiratory infections and create a substantial burden of disease, particularly in young children (<5 years), older individuals (≥65 years) and immune-compromised patients. HMPV has been detected in 4–16% of hospitalizations of children suffering from ARIs. Serologic surveys revealed that primary HMPV infection is common, as antibodies were detected in over 90% of individuals between 5 and 10 years of age [2,3].

HMPV has been isolated on all continents and has a seasonal distribution. Outbreaks occur mainly in the winter and spring months in the northern hemisphere and between June and July in the southern hemisphere. [4,5]. HMPV is a recent virus of the genus Metapneumovirus, belonging to the family Pneumoviridae. It was first isolated and identified in 2001 in the Netherlands in nasopharyngeal aspirates from children and adults with acute respiratory tract infections. HMPV hospitalization rates were the highest among children <2 years of age (200 of 100,000 person-year) and is lower among children between 5 and 17 years of age (5 of 100,000). HMPV has been associated with symptoms ranging from mild upper respiratory tract infections (cough, fever, wheezing, rhinorrhea, dyspnea, cyanosis, myalgia, vomiting, diarrhea) and more severe ones (bronchiolitis and pneumonia) [6,7,8].

In Mexico, influenza surveillance is conducted by the national epidemiological surveillance system through sentinel monitoring. Influenza monitoring health units capture all probable cases of influenza on the national influenza platform and only take 10% of samples of patients with influenza-like diseases and 100% samples of patients with severe acute respiratory Infection. These health units are distributed throughout the country and include different sectors within the Ministry of Health (National Institute of Social Security for State workers, Mexican Institute of Social Security, Petroleos Mexicanos, Secretary of Defense, Secretary of the Navy). The surveillance of non-influenza respiratory viruses is conducted in the country based on samples of hospitalized patients and deaths that meet the operational definition of severe acute respiratory infection. From 2009 to 2018, 188,076 hospitalized cases and 16,812 deaths were registered on the national influenza platform, of which 111,260 and 8586, respectively, tested negative for influenza. Of the 119,846 samples of hospitalized patients and deaths throughout the health sector, 7283 (6%) samples were processed with the objective of identifying respiratory etiologic agents and determining the epidemiological behavior of HMPV in influenza-negative samples from hospitalized and deceased patients in Mexico between 2009 and 2018. The results of this study aim to raise awareness among clinicians about the impact of HMPV on severe acute respiratory infections, mainly among young children and older people. 

## 2. Results

### 2.1. Epidemiology of HMPV

A total of 7283 samples were collected from all over the country from patients with SARI between April 2009 and December 2018. Out of the samples, 40.8% (2973/7283) of the influenza-negative ones were positive for respiratory viruses. Viruses with the highest circulation were rhinovirus/enterovirus (HRV/HEV) (47.6%), respiratory syncytial virus 15.9% (RSV) (RSV A 7.8%, RSV B 6.9% and RSV 1.2%, subgroup identification was performed by real-time RT-PCR), metapneumovirus (HMPV) (11.1%), parainfluenza 3 (PIV3) (5.1%), adenovirus (AdV) (3.0%), co-infections (10.8%), parainfluenza 1 (PIV1) (2.0%), parainfluenza 2 (PIV2) (0.6%), parainfluenza 4 (PIV4) (1.2%), coronavirus (CoV 229E) (0.9%), CoV HKU1 (0.4%), CoV NL63 (0.8%), CoV OC43 (0.5%) and Bocavirus (BoV) (0.1%). In total, 59% of the samples were negative, and 0.2% were undetermined (Figure 1).

A total of 57.4% (190/331) of the positive samples of HMPV were pharyngeal swab, 39% (129/331) were nasopharyngeal swab, 3.3% (11/331) were bronchioalveolar washes and 0.3% (1/331) were biopsies. 89% of the HMPV positive samples (297/331) were taken during the first five days of the onset of symptoms, and 10.3% (34/331) were taken six to 10 days after the onset of symptoms. In females, there were 55% of positive cases of HMPV, while this was 45% in males.

#### 2.1.1. Age Groups Affected by HMPV

HMPV was present in all age groups, but the most affected age groups were individuals between 29 days and 12 months (28.1%), of 13–24 months (15.1%), of 25–48 months (12.7%), of 5–9 years (9.7%), of 40–64 years (12.7%) and of more than 65 years (13%). The lowest percentage of circulation was found in neonates from 0 to 28 days (1.5%) and individuals from 10 to 39 years (7.3%). The analysis of the age groups in confections with HMPV showed the same pattern. A higher percentage of positivity was present in the age group from 29 days to 12 months, followed by the age group from 13 months to 9 years. The rest of the age groups were affected but to a smaller degree (Figure 2a,b).

The age groups and genders of all the samples included in this study were compared to positive HMPV cases. It was observed that severe acute respiratory infections had a greater percentage among the age group from 29 days to 12 months. HMPV positive samples also predominantly belonged to this age group. Males and females presented with 51.3% (3739) and 48.7% (3544), respectively, of the total samples. For the HMPV positive samples, 45% (149) were males and 55% (182) were females (Table 1).

#### 2.1.2. Seasonal Distribution of HMPV

In Mexico, HMPV has a marked seasonality, which begins in November and ends in April. The highest peak of activity has been observed to be at the end of winter (January and February) (Figure 3).

#### 2.1.3. Geographical Distribution of HMPV

Mexico has 32 states, which are divided into four zones: the north zone (Baja California, Baja California Sur, Chihuahua, Coahuila, Durango, Nuevo Leon, Sinaloa, Sonora, Tamaulipas and Zacatecas), center zone (Aguascalientes, Colima, Ciudad de Mexico, Guanajuato, Hidalgo, Jalisco, Mexico, Michoacan, Morelos, Nayarit, Queretaro, San Luis Potosi, Tlaxcala), southeast zone (Chiapas, Guerrero, Oaxaca, Puebla, Tabasco, Veracruz) and peninsula zone (Campeche, Quintana Roo and Yucatan). The circulation of HMPV was observed predominantly in the states of the north and the center zones. The states of the southeast and the peninsula zones showed less circulation. The same pattern was observed when comparing age groups to zones (Figure 4 and Figure 5).

### 2.2. Other Respiratory Viruses Coinfecting with HMPV

Co-infections with other respiratory viruses were shown by 10% of the positive samples (320/2973), and 24.7% (79/320) were co-infections with HMPV. The most frequent co-infections were HMPV/HRV/HEV (Table 2).

### 2.3. Clinical Manifestations of HMPV Infection

The diagnosis of other respiratory viruses negative to influenza was only made in hospitalized and deceased patients. Table 3 presents the classification of hospitalized patients at the time of the sample process, it is important to mention that the status of hospitalized patients with severe acute respiratory infection is not always updated by clinicians, thus, it is unknown if non-serious patients (not intubated with severe acute respiratory infection) evolved to severe patients (intubated with pneumonia) or if severe patients left the hospital for improvement or died. 

The most frequent symptoms in the positive cases of HMPV were cough, fever, dyspnea, attack on the general condition, sudden onset of symptoms, rhinorrhea, irritability, polypnea and headache. The rest of the symptoms presented in a smaller proportion. The most frequent comorbidities in positive cases of HMPV were diabetes, hypertension, chronic obstructive pulmonary disease, obesity, asthma, smoking and cardiovascular disease; comorbidities such as immunosuppression, HIV and chronic renal failure occur in a smaller proportion (Table 4).

It was observed that 21% of the comorbidities occurred in age groups under 34 years. HIV (80%), immunosuppression (67%), asthma (65%) and cardiovascular disease (50%) were the comorbidities with the highest percentage in this age group. Among the age groups older than 35 years, 79% presented renal failure (100%), hypertension (97%), chronic obstructive pulmonary disease (94%), smoking (92%), obesity (91%), diabetes (90%) and cardiovascular disease (50%). The age group from 65 to 69 was the one with the highest number of comorbidities. It was also observed that when there were a lower number of comorbidities, the percentage of HMPV positivity was higher, while when there were a greater number of comorbidities, the percentage of positivity for HMPV was lower (Figure 6).

## 3. Discussion

The most common respiratory viruses causing ARI were RSV, HMPV, PIV and HRV. These results are consistent with the viruses identified in this study [9,10]. Studies in hospitalized and outpatient pediatric patients worldwide have reported that HMPV is associated with 6–40% of acute respiratory illness [11].

The percentage of HMPV positivity was 11.1%. This is comparable with several studies that mentioned that HMPV positivity varies widely from as low as 1.7% to as high as 17%, with generally higher prevalence in children younger than five years compared to older age groups [12,13,14,15].

Females were the most affected, according to one study; the female gender is a risk factor for developing a severe disease due to HMPV [16]. Studies show that the highest prevalence of HMPV occurs in females [17].

One study reported that HMPV positive cases were identified within four days of the onset of symptoms, and another study mentioned that peak viral titers are seen between days four and five in mice and cotton rats during animal experimentation. This is similar to our study where we found that 89.5% of positive samples of HMPV were collected within the first 5 days of the onset of symptoms [6]. 

Our findings showed that HMPV infections occurred in all age groups, but the most affected age groups were those under five years of age, mainly infants less than one year old and older people. These results are consistent with the serological studies, which suggest that HMPV is acquired early in life, and at five years of age, approximately 70% develop antibodies against HMPV. The prevalence of antibodies against HMPV in people between the ages of six months and one year is 25%, and this prevalence increases to 55% for those between one and two years, to 70% for those from two to five years and 100% for individuals more than five years of age [18,19]. 

One study reported that approximately 85–100% of the adult population has antibodies against HMPV, and it is likely that these preexisting antibodies are not completely protective. Insufficient immunity acquired during initial infection and/or due to infections from different genotypes does not cause cross-protection immunity. All of this suggests that re-infection throughout life may be common [20,21,22]. Despite the fact that HMPV infection occurs mildly in healthy younger adults, infection with HMPV increases disease severity and high morbidity and mortality rates in the elderly [11].

HMPV epidemics occurred at the end of winter and spring in most temperate sites, but the timing of epidemics was more diverse in the tropics. The duration of epidemics was 4.8 months (4.4 to 5.1) [2]. In Mexico, from 2009 to 2018, the circulation of HMPV began in November and ended in April. Thus, the duration of the HMPV season was approximately six months. 

In Mexico, the circulation of HMPV was observed mainly in the northern and the central areas of the country where winter temperatures fluctuated between 5 °C and 7 °C, while in the areas of the southeast and the peninsula where the circulation of HMPV is lower, winter temperatures are higher (14 to 17 °C). The predominance of HMPV in the states with the lowest temperature in winter is consistent with a study that mentions that HMPV sees an increase in winter is due to the fact that the decrease in air temperature causes a decrease in nasal airway temperature (compromised cooling of the nasal airway), which leads to a poor defense against infection [9]. 

A study reported that viral co-infection rates in patients with HMPV range from 6% to 23% [11]. In this study, we found HMPV in co-infection with other respiratory viruses, mainly HRV/HEV, followed by CoV, PIV, AdV, RSV and BoV. There are different studies that mention that HMPV and RSV, two common pathogens, overlap at the same time [23]. Another study reported that during the influenza A (H1N1)pdm09 epidemic, influenza-like illness in children under 5 was caused by influenza viruses (25%), RSV (19%), HRV (17%), HMPV (9%) and PIV (7%). All these viruses circulated at the same time [24]. Another study stated that HRV is responsible for the majority of common colds in winter and occurs in co-infection with other respiratory viruses (influenza, RSV, HMPV, PIV, CoV, BoV and AdV), and the proportion in viral respiratory co-infection was similar between HEV and HRV [25]. 

There are studies that describe the main symptoms in positive cases: cough, fever, dyspnea, sore throat, hoarseness of voice, rhinorrhea, myalgia, anorexia, vomiting, irritability, pharyngitis, otitis media, conjunctivitis, diarrhea and lethargy. These results align with the findings of this study [16,17,26]. 

HMPV infection may be more severe in patients with underlying medical conditions. It has been shown that 30–85% of children hospitalized with HMPV have chronic conditions such as asthma, chronic lung disease due to prematurity or congenital heart disease. In addition, HMPV is an important cause of acute respiratory disease in older adults (>65 years) and adults with comorbid diseases such as chronic obstructive pulmonary disease, asthma, immunosuppression and chronic renal failure. Much of the comorbidity was based on positive HMPV, mainly in people over 40 years of age [18]

## 4. Materials and Methods 

### 4.1. Clinical Samples.

The respiratory samples (pharyngeal swabs, nasopharyngeal swabs, bronchio-alveolar washes and biopsies) included in this study met the operational definition of Severe Acute Respiratory Infection (SARI) established by the World Health Organization [27]. The samples were collected by the Influenza Network of Monitoring Health Units from all across the 32 states of the Mexican Republic. These were taken with the help of rayon or dacron swabs with plastic tips in the case of pharyngeal swabs and with flexible tips for nasopharyngeal swabs. In the case of biopsies, 2 cm^2^ of lung parenchyma were collected from the most affected area. The samples were placed immediately in a viral transport medium and handled from 2 to 8 °C prior to transportation to the laboratory. Clinical and epidemiological information was captured on the influenza platform of the National Epidemiological Surveillance System.

The samples were processed with the help of RT-PCR in real time in the National Influenza Laboratory Network all across the 32 states of the Mexican Republic. The samples with negative results for influenza in hospitalized and deceased patients were sent for the diagnosis of other respiratory viruses to the Institute of Epidemiological Diagnosis and Reference “Dr. Manuel Martinez Baez” (InDRE), recognized as the National Influenza Center in Mexico by the World Health Organization. 

This study included 7283 SARI influenza-negative respiratory samples from hospitalized and deceased patients for the diagnosis of other respiratory viruses. These samples were processed by qualitative multiplex RT-PCR for the simultaneous detection of nucleic acids from 14 respiratory viruses (xTAG^®^ RVP FAST v2 with the universal classification system of Luminex Molecular Diagnostics on Luminex^®^ Platforms LX100 / 200) between April 2009 and December 2018.

### 4.2. Extraction of Nucleic Acids

Total viral nucleic acid was extracted from 200 µL of the respiratory samples and 20 µL of Lamda bacteriophage used as the internal control (MS2 for xTAG^®^). The extraction was carried out with the QIAamp^®^MinElute^®^Virus Spin kit (QIAGEN) according to the manufacturer’s specifications. Nucleic acids were stored at 4 °C when processed the same day or stored at −80 °C for later analysis.

### 4.3. Qualitative Multiplex RT-PCR (xTAG^®^ RVP FAST v2)

We used a qualitative multiplex test for the simultaneous detection and identification of nucleic acids of different respiratory viruses (RSV, HMPV, HRV / HEV, PIV1, PIV2, PIV3, PIV4, AdV, HCoV HKU1, HCoV OC43, HCoV NL63, HCoV 229E and HBoV) from respiratory samples. The xTAG^®^ RVP FAST v2 test incorporates the multiplex reverse transcription polymerase chain reaction (RT-PCR) in Luminex platforms together with the universal tag classification system. The analysis also detects the MS2 bacteriophage, which must be added to each sample as an internal control and a procedure control prior to extraction (lamda bacteriophage DNA).

In total, 10 µL of the extract (RNA or DNA) of each sample was used to perform the multiplex RT-PCR. For each virus or internal control present in the sample, PCR amplifiers were obtained, with sizes ranging from 58 to 206 bp, not including the 24-segment tag. Next, 2 µL of the RT-PCR product is added to a hybridization/detection reaction containing a mixture of microspheres and the streptavidin/R-phycoerythrin conjugate. Each population of Luminex microspheres detects a highly specific viral target or analysis control by means of a specific anti-tag/tag hybridization. After incubation of the RT-PCR products with the microsphere mixture, the Luminex instrument classifies and reads the hybridization/detection reactions. A medium fluorescence intensity (MFI) signal is generated for each population of microspheres (viral target or analysis control). Fluorescence values are analyzed in order to determine the presence or absence of virus in each sample analyzed.

The data generated by the Luminex instrument is analyzed with the help of the xTAG Data Analysis Software for Respiratory Viral Panel Fast v2 (TDAS RVP FAST) in order to generate a report that summarizes the viruses present in the sample [28].

### 4.4. Real time RT-PCR (Detection of RSV A and B)

In total, 5 µL of the extract (RNA) from each sample of RSV was used to perform the real time RT-PCR. The primers and probes used included RSV A Forward (5′-GCTCTTAGCAAAGTCAAGTTGAATGA-3′), RSV A Reverse (5′-TGCTCCGTTGGATGGTGTATT-3′), RSV A Probe (5′-6FAM ACACTCAACAAAGATCAACTTCTGTCATCCAGC TAMRA-3′), RSV B Forward (5′-GATGGCTCTTAGCAAAGTCAAGTTAA-3′), RSV B Reverse (5′-TGTCAATATTATCTCCTGTACTACGTTGAA-3′) and RSV B Probe (5′-6FAM TGATACATTAAATAAGGATCAGCTGCTGTCATCCA TAMRA-3′). Real time RT-PCR was perform using SuperScript III Platinum One-Step qRT-PCR kit (Thermo Fisher Scientific) with the following conditions: reverse transcription at 50 °C for 30 min (1 cycle); Taq polymerase activation at 95 °C for 5 min (1 cycle), PCR amplification 95 °C for 15 s and 55 °C for 30 s (45 cycles). The thermal cycling was conducted in the 7500 Fast Real-Time PCR System (Applied Biosystem).

## 5. Conclusions

This study has limitations because it does not aim at the intentional search for HMPV; other respiratory viruses were surveilled from a small percentage of hospitalized patients and deaths with severe acute respiratory infection infected with influenza throughout the country with the objective of providing epidemiological information on circulating non-influenza respiratory viruses in Mexico.

In conclusion, our study shows that severe acute respiratory infections in hospitalized and deceased patients from influenza-negative samples are mainly caused by rhinovirus/enterovirus, respiratory syncytial virus, metapneumovirus and parainfluenza. All of these pathogens share symptomatology and seasonality. For this reason, it is important to have a better knowledge of each of them. 

The generation of information about HMPV with respect to clinical manifestations, affected age groups, co-infections with other respiratory viruses, geographical distribution and seasonality is important for planning and development public health policies for adequate clinical management, reduction of the indiscriminate use of antibiotics in the context of the increased antimicrobial resistance worldwide and decrease of the economic impact in terms of lost workdays, lost school days and additional costs of medical care.

## Figures and Tables

**Figure 1 pathogens-09-00085-f001:**
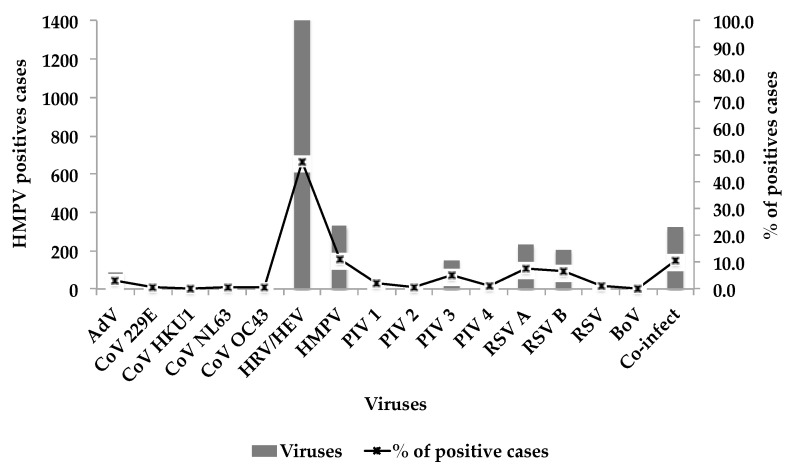
Distribution of respiratory viruses in influenza-negative samples from hospitalized and deceased patients detected by qualitative multiplex RT-PCR (xTAG^®^ RVP FAST v2) from April 2009 to December 2018 in Mexico.

**Figure 2 pathogens-09-00085-f002:**
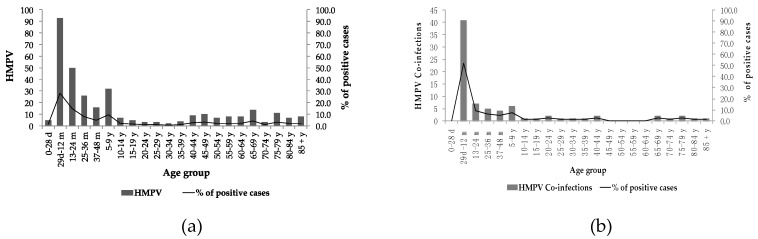
(**a**) Age group distribution of HMPV positive cases and % of positive cases; (**b**) Age group distribution of HMPV co-infections and % of positive cases.

**Figure 3 pathogens-09-00085-f003:**
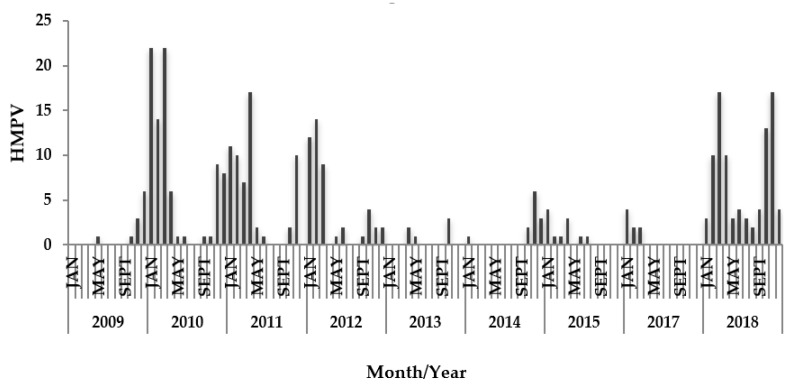
The monthly/annual distribution of HMPV in SARI hospitalized and deceased patients between April 2009 and December of 2018 in Mexico.

**Figure 4 pathogens-09-00085-f004:**
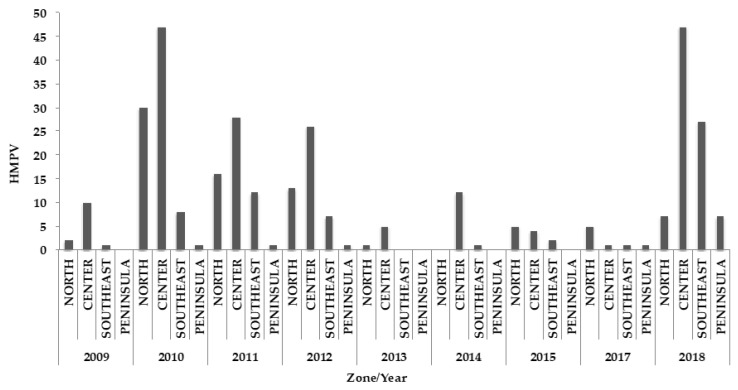
Annual distribution of HMPV by zones: north, center, southeast and peninsula.

**Figure 5 pathogens-09-00085-f005:**
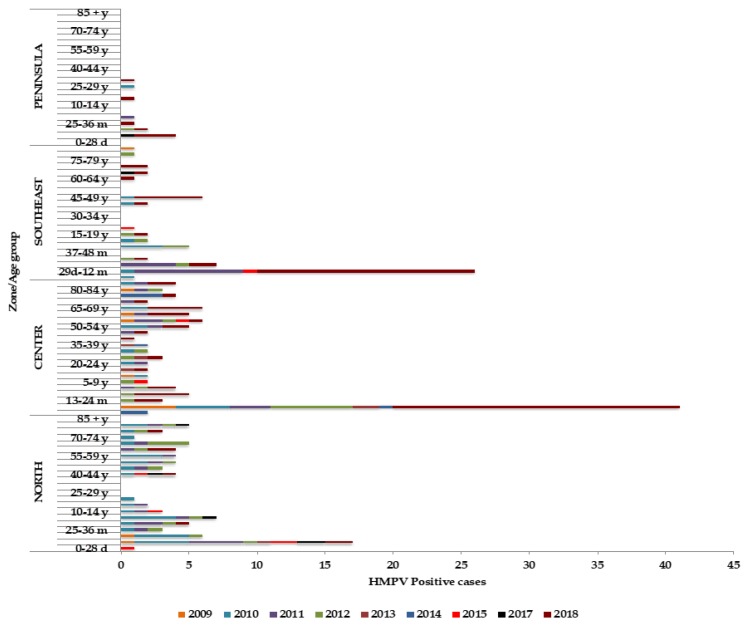
Annual distribution of HMPV by zones: north, center, southeast and peninsula and age group.

**Figure 6 pathogens-09-00085-f006:**
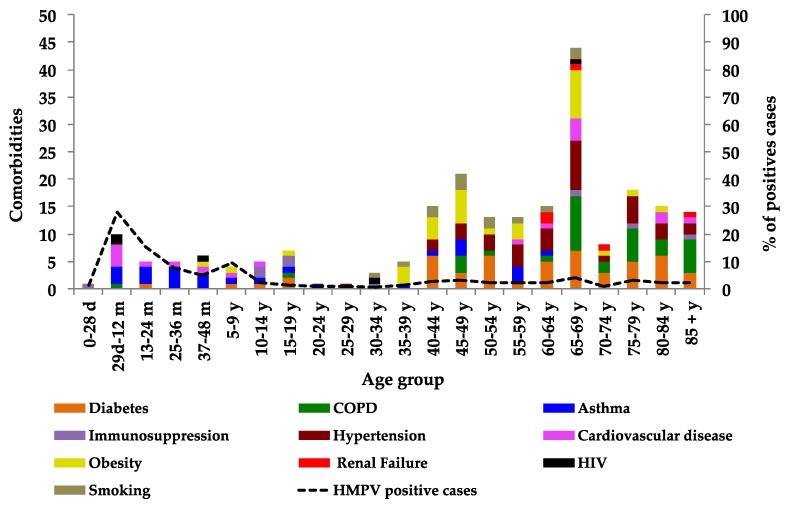
Distribution of comorbidities by age group and percentage of positive cases.

**Table 1 pathogens-09-00085-t001:** Comparison of the age group and gender of the samples included in this study and the HMPV positive samples.

Age Group	All Samples			Positives		
	Total Samples	Male	Female	HMPV	Male	Female
0–28 d	233 (3.2%)	148 (4.0%)	85 (2.4%)	5 (1.5%)	3 (2.0%)	2 (1.1%)
29d–12 m	1737 (23.9%)	1004 (26.9%)	733 (20.7%)	93 (28.1%)	58 (39%)	35 (19.2%)
13–24 m	287 (3.9%)	146 (3.9%)	141 (4.0%)	50 (15.2%)	22 (14.8%)	28 (15.4%)
25–36 m	196 (2.7%)	96 (2.6%)	100 (2.8%)	26 (7.9%)	13 (8.7%)	13 (7.1%)
37–48 m	157 (2.1%)	84 (2.2%)	73 (2.1%)	16 (4.8%)	3 (2.0%)	13 (7.1%)
5–9 y	325 (4.4%)	175 (4.7%)	150 (4.2%)	32 (9.8%)	11 (7.4%)	21 (11.6%)
10–14 y	198 (2.7%)	107 (2.9%)	91 (2.6%)	7 (2.1%)	3 (2.0%)	4 (2.2%)
15–19 y	241 (3.3%)	106 (2.8%)	135 (3.8%)	5 (1.5%)	3 (2.0%)	2 (1.1%)
20–24 y	285 (3.9%)	128 (3.4%)	157 (4.4%)	3 (0.9%)	1 (0.7%)	2 (1.1%)
25–29 y	359 (4.9%)	165 (4.4%)	194 (5.5%)	3 (0.9%)	1 (0.7%)	2 (1.1%)
30–34 y	328 (4.5%)	146 (3.9%)	182 (5.1%)	2 (0.6%)	0 (0%)	2 (1.1%)
35–39 y	335 (4.6%)	152 (4.1%)	183 (5.2%)	4 (1.2%)	2 (1.3%)	2 (1.1%)
40–44 y	308 (4.2%)	141 (3.8%)	167 (4.7%)	9 (2.7%)	2 (1.3%)	7 (3.8%)
45–49 y	288 (4.0%)	140 (3.7%)	148 (4.2%)	10 (3.0%)	3 (2.0%)	7 (3.8%)
50–54 y	276 (3.8%)	149 (4.0%)	127 (3.6%)	7 (2.1%)	2 (1.3%)	5 (2.7%)
55–59 y	253 (3.5%)	126 (3.4%)	127 (3.6%)	8 (2.4%)	2 (1.3%)	6 (3.3%)
60–64 y	261 (3.6%)	132 (3.5%)	129 (3.6%)	8 (2.4%)	5 (3.4%)	3 (1.7%)
65–69 y	212 (2.9%)	105 (2.8%)	107 (3.0%)	14 (4.2%)	5 (3.4%)	9 (4.9%)
70–74 y	218 (3.1%)	108 (2.9%)	110 (3.1%)	3 (0.9%)	1 (0.7%)	2 (1.1%)
75–79 y	180 (2.5%)	89 (2.4%)	91 (2.6%)	11 (3.3%)	3 (2.0%)	8 (4.4%)
80–84 y	175 (2.4%)	91 (2.4%)	84 (2.4%)	7 (2.1%)	3 (2.0%)	4 (2.2%)
85 + y	257 (3.5%)	118 (3.2%)	139 (3.8%)	8 (2.4%)	3 (2.0%)	5 (2.8%)
No data	174 (2.4%)	83 (2.2%)	91 (2.6%)	0 (0%)	0 (0%)	0 (0%)
Total	7283 (100%)	3739 (51.3%)	3544 (48.7%)	331 (100%)	149 (45%)	182 (55%)

The data is shown as a number (%); d (days), m (months) and y (years of age).

**Table 2 pathogens-09-00085-t002:** Distribution of positive cases and co-infections of HMPV.

Positive Cases	HMPV
HMPV	331
HMPV/HRV/HEV	40
HMPV/PIV3	7
HMPV/HCoV HKU1	8
HMPV/AdV	5
HMPV/HCoV 229E	4
HMPV/PIV1	3
HMPV/HCoV OC43	2
HMPV/BoV	1
HMPV/RSVA	1
HMPV/RSVB	1
HMPV/A(H1N1)pdm09	1
HMPV/HRV/HEV/RSV	1
HMPV/HRV/HEV/PIV1	1
HMPV/HRV/HEV/PIV3	1
HMPV/HCoV 229E/HRV/HEV	1
HMPV/RSV/PIV1	1
HMPV/RSV/AdV	1
Total	410

The data is shown as a number.

**Table 3 pathogens-09-00085-t003:** Status of patients positive to HMPV.

Hospitalized Status	HMPV
Severe case	125
Not Severe case	120
High for improvement	59
Death	27
Total	331

The data is shown as a number.

**Table 4 pathogens-09-00085-t004:** Clinical manifestation and co-morbidities in positive cases of HMPV.

Symptoms	HMPV	Comorbidities	HMPV
Cough	322/331 (97.3%)	Diabetes	52/331 (15.7%)
Fever	315/331 (95.2%)	Hypertension	39/331 (11.8%)
Dyspnea	306/331 (92.4%)	Chronic obstructive pulmonary disease	34/331 (10.3%)
Attack to the general condition	306/331 (92.4%)	Obesity	32/331 (9.7%)
Sudden onset of symptoms	286/331 (86.4%)	Asthma	26/331 (7.8%)
Rhinorrhea	257/331 (77.6%)	Smoking	22/331 (6.6%)
Irritability	234/331 (70.7%)	Cardiovascular Disease	19/331 (5.7%)
Polypnea	224/331 (67.7%)	Immunosuppression	9/331 (2.7%)
Headache	191/331 (57.7%)	HIV	5/331 (1.5%)
Shaking chills	153/331 (46.2%)	Chronic renal failure	5/331 (1.5%)
Odynophagia	130/331 (39.3%)		
Muscle ache	130/331 (39.3%)		
Arthralgia	118/331 (35.6%)		
Thoracic pain	116/331 (35.0%)		
Vomit	82/331 (24.8%)		
Conjunctivitis	81/331 (24.5%)		
Cyanosis	76/331 (22.9%)		
Diarrhea	58/331 (17.5%)		
Abdominal pain	51/331 (15.4%)		

The data is shown as a number and percentage (%).

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
