# Peer review of "Human Metapneumovirus: Etiological Agent of Severe Acute Respiratory Infections in Hospitalized and Deceased Patients with a Negative Diagnosis of Influenza"

_pathogens, 2020, doi:10.3390/pathogens9020085_

Round 1

Reviewer 1 Report

This manuscript describes a retrospective epidemiologic study concerning non-influenza respiratory viruses, and especially the human metapneumovirus, responsible of severe acute respiratory infections over the last decade (2009-2018) in Mexico. The method used for the diagnostic are sound and the quantity and variety of the analysed samples, from all age groups, is relevant for this study. However, the design of the study exploring  HMPV epidemiology in terms of number of validated HMPV-related SARI cases, age groups, regional repartition in Mexico, seasonality, co-infections and comorbidities is very common.

Major comments concerning this manuscript :

The global English level need to be improved to ease the comprehension of the results and their interpretation. Some sentences have syntax issues, typing errors. Spanish words are included in the text at several places. HMPV is typed HMVP a lot in the manuscript and figures.

Line 45-46 : the taxonomy of HMPV is wrong. The Pneumoviridae is a family apart of the Paramyxoviridae. “Recent” virus instead of “new” virus would be more appropriate for a virus discovered in 2001.

Line 62 : “Respiratory Syncytial Virus 15.9% (RSV) (RSV A 7.8%, RSV B 6.9% and RSV 1.2%),…” Is it true that RSV A and B are distinctly detected by the diagnostic kit used (xTAG® RVP FAST v2) ? If it is, what are the 1,2% attributed to RSV ? In parallel, further considerations about HMPV subgroups, their distribution and circulation over the years covered in this study would be of great interest.

The quality of Figure 4 and especially Figure 5 is insufficient for its reading. It is not clear if the co-infections HMPV with other respiratory viruses are represented in these figures and no description or/interpretation is made in the text. Moreover, if it is correct that co-infections data are included in these figures, the actual representation is not adapted to see it.

No statistical analysis has been made in this study. For example, is the prevalence of HMPV in the female gender significantly higher than in the male gender in your study ?

Lines 126-128, “comorbidities such as asthma and cardiovascular disease occurred in age groups under 19 years, while the rest of the comorbidities of cases positive to HMVP were presented after 35 years of age”. The colours chose in the graph are very similar but it seems that asthma and cardiovascular disease are comorbidities in all age groups and not only under 19 y-o. Can you explain your conclusion ?

In figure 6, there is an error of title. Moreover, what represents the kind-of grey colour present in the graph in groups between 30 and 69 y-o ? there is no such a colour in the caption.

Minor comments :

In figures 1, 2,6, “% Positivity” should be replaced by “% of positives cases”. Figure 2b, symbols appeared after axis descriptions. Line 19 : This study appeared to be more a retrospective study than a prospective one.

Author Response

Major comments concerning this manuscript :

The global English level need to be improved to ease the comprehension of the results and their interpretation. Some sentences have syntax issues, typing errors. Spanish words are included in the text at several places. HMPV is typed HMVP a lot in the manuscript and figures.

To make it better, we have modified your diction and sentence structures. Moreover, the grammar, punctuation, spelling, usage, and consistency errors in your document have been corrected. Further, the clarity, cohesion, syntax, and more have been refined where necessary.

Line 45-46 : the taxonomy of HMPV is wrong. The Pneumoviridae is a family apart of the Paramyxoviridae. “Recent” virus instead of “new” virus would be more appropriate for a virus discovered in 2001.

Line 45-46 :HMPV is a recent virus of the genus Metapneumovirus belonging to the family Pneumovirinae

Line 62 : “Respiratory Syncytial Virus 15.9% (RSV) (RSV A 7.8%, RSV B 6.9% and RSV 1.2%),…” Is it true that RSV A and B are distinctly detected by the diagnostic kit used (xTAG® RVP FAST v2) ? If it is, what are the 1,2% attributed to RSV ? In parallel, further considerations about HMPV subgroups, their distribution and circulation over the years covered in this study would be of great interest.

Line 62: The xTAG® RVP FAST v2 does not directly detect RSV A and B, the identification of subgroup A and B was performed using real-time RT-PCR and 1.2% failed to identify. We are currently designed primers to genotype by sequencing the HMPV positive samples from 2009 to 2019, but we will obtain these results in 2020.

Line 64. We add “subgroup identification was performed by real-time RT-PCR”

Line 277-289. We add “4.4. Real time RT-PCR (Detection of RSV A and B)

5 µL of the extract (RNA) of each sample of RSV were used to perform the real time RT-PCR. The primers and probes used were RSV A Forward (5´-GCTCTTAGCAAAGTCAAGTTGAATGA-3´), RSV A Reverse (5´-TGCTCCGTTGGATGGTGTATT-3´), RSV A Probe (5´-6FAM ACACTCAACAAAGATCAACTTCTGTCATCCAGC TAMRA-3´), RSV B Forward (5´-GATGGCTCTTAGCAAAGTCAAGTTAA-3´), RSV B Reverse (5´-TGTCAATATTATCTCCTGTACTACGTTGAA-3´), RSV B Probe (5´-6FAM TGATACATTAAATAAGGATCAGCTGCTGTCATCCA TAMRA-3´). Real time RT-PCR was perform using SuperScript III Platinum One-Step qRT-PCR kit (Thermo Fisher Scientific) with the following conditions: reverse transcription at 50ºC for 30 min (1 cycle); Taq polymerase activation at 95ºC for 5 min (1 cycle), PCR amplification 95ºC for 15 seconds and 55ºC for 30 seconds (45 cycles). The thermal cycling was carried out in the 7500 Fast Real-Time PCR System (Applied Biosystem).

The quality of Figure 4 and especially Figure 5 is insufficient for its reading. It is not clear if the co-infections HMPV with other respiratory viruses are represented in these figures and no description or/interpretation is made in the text. Moreover, if it is correct that co-infections data are included in these figures, the actual representation is not adapted to see it.

 Graphs 4 and 5 were changed to improve visualization. Only those positive for Metapneumovirus were included in the graphs, coinfections were excluded because they are not mentioned in the text.

No statistical analysis has been made in this study. For example, is the prevalence of HMPV in the female gender significantly higher than in the male gender in your study ?

 A statistical analysis was not carried out to determine if the female gender over the male gender is significant because the number of Metapneumovirus positive samples is small. Only the results obtained were described

Lines 126-128, “comorbidities such as asthma and cardiovascular disease occurred in age groups under 19 years, while the rest of the comorbidities of cases positive to HMVP were presented after 35 years of age”. The colours chose in the graph are very similar but it seems that asthma and cardiovascular disease are comorbidities in all age groups and not only under 19 y-o. Can you explain your conclusion ?

 Change this

Lines 126-128, “In this study, it was observed that comorbidities such as asthma, and cardiovascular disease occurred in age groups under 19 years, while the rest of the comorbidities of cases positive to HMVP were presented after 35 years of age”,

By

Lines 135-139, It was observed that 21% of the comorbidities occurred in age groups under 34 years. HIV (80%), immunosuppression (67%), asthma (65%) and cardiovascular disease (50%) were the comorbidities with the highest percentage in this age group. 79% of the age groups older than 35 years presented with renal failure (100%), hypertension (97%), chronic obstructive pulmonary disease (94%), smoking (92%), obesity (91%), diabetes (90%) and cardiovascular disease (50%).

In figure 6, there is an error of title. Moreover, what represents the kind-of grey colour present in the graph in groups between 30 and 69 y-o ? there is no such a colour in the caption.

The colors in Figure 6 were changed.

Minor comments :

In figures 1, 2,6, “% Positivity” should be replaced by “% of positives cases”. Figure 2b, symbols appeared after axis descriptions. Line 19 : This study appeared to be more a retrospective study than a prospective one.

Figures 1, 2 and 6 were changed by “% of positives cases”.

Line 19 : We changed prospective by This retrospective study

Reviewer 2 Report

This is a manuscript describing a retrospective study using convenience samples collected from patient with acute respiratory infection. The study is of interest due to the large number of samples, national scope, multiple seasons, and a country for which there is limited data on HMPV. The results show very interesting seasonal and geographic variation. However, there are significant limitations to the work. Some of the limitations are due to the nature of the study, but others could be addressed to strengthen the conclusions and value to the field.

Line 46. The Pneumoviridae are now a separate family, rather than a subfamily of the Paramyxoviridae. Results, lines 59-60 and Methods, lines 200-202. Is there any data on what percentage of all patients are represented here? The text states “total of 7283 samples were collected in all the country from patients with SARI since April 2009 to December 2018”. Presumably, given the population of Mexico, this represents a very small fraction of the total SARI hospitalizations during that time period. It would provide context if the authors were able to estimate how representative this sample is. The authors should show a table of the age distribution and gender of all the samples, not only the positives. This is important to provide context to the age distribution and gender of the HMPV-positive patients. Results, figure 4 and 5. Figure 4 is interesting because it shows the seasonal and regional variability in rates. I’m not sure what Figure 5 adds, plus it is very small and hard to read. Results, Table 2. This is difficult to interpret. What was the definition of severe? What does “high for improvement” mean? If the authors could compare the severity of HMPV to, for example, the severity of influenza in the same dataset, that would be very valuable. As is, the data simply indicate that some HMPV cases are severe and even fatal, but this is known. Discussion, lines 142-144. This is an incomplete citing of the literature. The majority of studies of HMPV (like RSV and most respiratory viruses) shower higher rates of severe disease in males, especially among children. There should be a paragraph discussing the limitations of the study (retrospective, convenience samples, limited clinical/demographic/epidemiologic data, etc.). There are numerous typos and misspellings; e.g., “HMVP”, other virus names capitalized, etc.

Author Response

This is a manuscript describing a retrospective study using convenience samples collected from patient with acute respiratory infection. The study is of interest due to the large number of samples, national scope, multiple seasons, and a country for which there is limited data on HMPV. The results show very interesting seasonal and geographic variation. However, there are significant limitations to the work. Some of the limitations are due to the nature of the study, but others could be addressed to strengthen the conclusions and value to the field.

Line 46. The Pneumoviridae are now a separate family, rather than a subfamily of the Paramyxoviridae.

Line 46-47: We changes “HMPV is a new virus of the genus Metapneumovirus, of the subgroup Pneumovirinae belonging to the family Paramixoviridae.” By “HMPV is a recent virus of the genus Metapneumovirus belonging to the family Pneumovirinae”

Results, lines 59-60

and Methods, lines 200-202.

Is there any data on what percentage of all patients are represented here? The text states “total of 7283 samples were collected in all the country from patients with SARI since April 2009 to December 2018”. Presumably, given the population of Mexico, this represents a very small fraction of the total SARI hospitalizations during that time period. It would provide context if the authors were able to estimate how representative this sample is.

Line 54-70: We added “In Mexico, influenza surveillance is conducted by the national epidemiological surveillance system through sentinel monitoring. Influenza monitoring health units capture all probable cases of influenza on the national influenza platform and only take 10% of samples of patients with influenza-like diseases and 100% samples of patients with severe acute respiratory Infection. These health units are distributed throughout the country and include different sectors within the Ministry of Health (National Institute of Social Security for State workers, Mexican Institute of Social Security, Petroleos Mexicanos, Secretary of Defense, Secretary of the Navy). The surveillance of non-influenza respiratory viruses is conducted in the country based on samples of hospitalized patients and deaths that meet the operational definition of severe acute respiratory infection. From 2009 to 2018, 188076 hospitalized cases and 16812 deaths were registered on the national influenza platform, of which 111260 and 8586, respectively tested negative for influenza. Of the 119846 samples of hospitalized patients and deaths throughout the health sector, 7283 (6%) samples were processed with the objective of identifying respiratory etiologic agents and determining the epidemiological behavior of HMPV in influenza-negative samples from hospitalized and deceased patients in Mexico between 2009 and 2018. The results of this study aim to raise awareness among clinicians about the impact of HMPV on severe acute respiratory infections, mainly among young children and older people”.

The authors should show a table of the age distribution and gender of all the samples, not only the positives. This is important to provide context to the age distribution and gender of the HMPV-positive patients.

The age groups and genders of all the samples included in this study were compared to positive HMPV cases. It was observed that severe acute respiratory infections had a greater percentage among the age group from 29 days to 12 months. HMPV positive samples also predominantly belonged to this age group. Males and females presented with 51.3% and 48.7%, respectively, of the total samples. For the HMPV positive samples, 45% were males and 55% were females (Table 1).

Table 1. Comparison of the age groups and gender of the samples included in this study and the HMPV positive samples

Age

Group

All samples

Positives

Total samples

Male

Female

HMPV

Male

Female

0-28 d

233 (3.2%)

148 (4.0%)

85 (2.4%)

5 (1.5%)

3 (2.0%)

2 (1.1%)

29d-12 m

1737 (23.9%)

1004 (26.9%)

733 (20.7%)

93 (28.1%)

58 (39%)

35 (19.2%)

13-24 m

287 (3.9%)

146 (3.9%)

141 (4.0%)

50 (15.2%)

22 (14.8%)

28 (15.4%)

25-36 m

196 (2.7%)

96 (2.6%)

100 (2.8%)

26 (7.9%)

13 (8.7%)

13 (7.1%)

37-48 m

157 (2.1%)

84 (2.2%)

73 (2.1%)

16 (4.8%)

3 (2.0%)

13 (7.1%)

5-9 y

325 (4.4%)

175 (4.7%)

150 (4.2%)

32 (9.8%)

11 (7.4%)

21 (11.6%)

10-14 y

198 (2.7%)

107 (2.9%)

91 (2.6%)

7 (2.1%)

3 (2.0%)

4 (2.2%)

15-19 y

241 (3.3%)

106 (2.8%)

135 (3.8%)

5 (1.5%)

3 (2.0%)

2 (1.1%)

20-24 y

285 (3.9%)

128 (3.4%)

157 (4.4%)

3 (0.9%)

1 (0.7%)

2 (1.1%)

25-29 y

359 (4.9%)

165 (4.4%)

194 (5.5%)

3 (0.9%)

1 (0.7%)

2 (1.1%)

30-34 y

328 (4.5%)

146 (3.9%)

182 (5.1%)

2 (0.6%)

0 (0%)

2 (1.1%)

35-39 y

335 (4.6%)

152 (4.1%)

183 (5.2%)

4 (1.2%)

2 (1.3%)

2 (1.1%)

40-44 y

308 (4.2%)

141 (3.8%)

167 (4.7%)

9 (2.7%)

2 (1.3%)

7 (3.8%)

45-49 y

288 (4.0%)

140 (3.7%)

148 (4.2%)

10 (3.0%)

3 (2.0%)

7 (3.8%)

50-54 y

276 (3.8%)

149 (4.0%)

127 (3.6%)

7 (2.1%)

2 (1.3%)

5 (2.7%)

55-59 y

253 (3.5%)

126 (3.4%)

127 (3.6%)

8 (2.4%)

2 (1.3%)

6 (3.3%)

60-64 y

261 (3.6%)

132 (3.5%)

129 (3.6%)

8 (2.4%)

5 (3.4%)

3 (1.7%)

65-69 y

212 (2.9%)

105 (2.8%)

107 (3.0%)

14 (4.2%)

5 (3.4%)

9 (4.9%)

70-74 y

218 (3.1%)

108 (2.9%)

110 (3.1%)

3 (0.9%)

1 (0.7%)

2 (1.1%)

75-79 y

180 (2.5%)

89 (2.4%)

91 (2.6%)

11 (3.3%)

3 (2.0%)

8 (4.4%)

80-84 y

175 (2.4%)

91 (2.4%)

84 (2.4%)

7 (2.1%)

3 (2.0%)

4 (2.2%)

85 + y

257 (3.5%)

118 (3.2%)

139 (3.8%)

8 (2.4%)

3 (2.0%)

5 (2.8%)

No data

174 (2.4%)

83 (2.2%)

91 (2.6%)

0 (0%)

0 (0%)

0 (0%)

Total

7283 (100%)

3739

3544

331

149

182

Results, figure 4 and 5. Figure 4 is interesting because it shows the seasonal and regional variability in rates. I’m not sure what Figure 5 adds, plus it is very small and hard to read.

We change the figure 5

Results, Table 2. This is difficult to interpret. What was the definition of severe? What does “high for improvement” mean? If the authors could compare the severity of HMPV to, for example, the severity of influenza in the same dataset, that would be very valuable. As is, the data simply indicate that some HMPV cases are severe and even fatal, but this is known.

This study was conducted with samples of hospitalized patients, within the hospitalized there are different patient status, those who remain serious, those who are not serious, those who are discharged for improvement and those who die

Discussion, lines 142-144. This is an incomplete citing of the literature. The majority of studies of HMPV (like RSV and most respiratory viruses) shower higher rates of severe disease in males, especially among children. There should be a paragraph discussing the limitations of the study (retrospective, convenience samples, limited clinical/demographic/epidemiologic data, etc.). There are numerous typos and misspellings; e.g., “HMVP”, other virus names capitalized, etc. 

Line 295-299. We add a paragraph ”This study has limitations because it does not aim at the intentional search for HMPV, other respiratory viruses were surveilled from a small percentage of hospitalized patients and deaths with severe acute respiratory infection infected with influenza throughout the country with the objective of providing epidemiological information on circulating non-influenza respiratory viruses in Mexico”.

Round 2

Reviewer 1 Report

As suggested, the authors had refined their manuscript to gain in clarity and to ease the reading. However, there is still several minor points to correct :

The spelling error of HMVP in place of HMPV is still present 8 times in the manuscript (text and tables), as soon as line 43.

"% positivity" in place of "% of positive cases" is still written in your figures 1, 2. Figure 6, please correct "positivity MPHV" in  figure 6 by "HMPV positive cases".

Figure 1 : Delete the symbol showing at the end of your axis titles. 

Figure 2 : Delete the symbol showing at the end of your axis title in 2(b), the legend beneath the graphs are truncated. 

Lines 106-108 : "Males and females presented with 51.3% and 48.7%, 106 respectively, of the total samples. For the HMPV positive samples, 45% were males and 55% 107 were females (Table 1)." It is not clear what is calculated as percentages in the Table 1. Also, the numbers of 45% male and 55% female cases of positives HMPV cases are not written anywhere. Please clarify your calculations.  

Figure 4 and 5 : Delete the symbol showing at the end of titles, axis and graduations.

"% positivity" in place of "% of positive cases" is still written in your figures 1, 2. Figure 6, please correct "positivity MPHV" in  figure 6 by "HMPV positive cases".

Author Response

The spelling error of HMVP in place of HMPV is still present 8 times in the manuscript (text and tables), as soon as line 43.

Change the HMVP by HMPV in all the manuscript (text, tables and figures)

"% positivity" in place of "% of positive cases" is still written in your figures 1, 2. Figure 6, please correct "positivity MPHV" in  figure 6 by "HMPV positive cases".

Change "% positivity" by "% of positive cases" in figures 1, 2.

Figure 6: We correct "positivity MPHV" by "HMPV positive cases".

Figure 1 : Delete the symbol showing at the end of your axis titles. 

Figure 2 : Delete the symbol showing at the end of your axis title in 2(b), the legend beneath the graphs are truncated. 

Figures 1 and 2 were modified so that they do not look truncated, an apology but I do not know what is the symbol showing at the end on the axis title, could you please specify what are the symbols.

Lines 106-108 : "Males and females presented with 51.3% and 48.7%, 106 respectively, of the total samples. For the HMPV positive samples, 45% were males and 55% 107 were females (Table 1)." It is not clear what is calculated as percentages in the Table 1. Also, the numbers of 45% male and 55% female cases of positives HMPV cases are not written anywhere. Please clarify your calculations.  

The percentages in the totals of the last row of table 1 were annexed

We change “Males and females presented with 51.3% and 48.7% respectively, of the total samples. For the HMPV positive samples, 45% were males and 55% were females (Table 1)”.

By “Males and females presented with 51.3% (3739) and 48.7% (3544), respectively, of the total samples. For the HMPV positive samples, 45% (149) were males and 55% (182) were females (Table 1)”.

Figure 4 and 5 : Delete the symbol showing at the end of titles, axis and graduations.

An apology for not making the modifications but could you please specify what are the symbols at the end of the titles, axis and graduations.

"% positivity" in place of "% of positive cases" is still written in your figures 1, 2. Figure 6, please correct "positivity MPHV" in  figure 6 by "HMPV positive cases".

Change "% positivity" by "% of positive cases" in figures 1, 2.

Figure 6: We correct "positivity MPHV" by "HMPV positive cases".

Reviewer 2 Report

The authors have responded nicely to most of my prior concerns. The clarification of the surveillance system is very helpful, as is the data on age distribution. The new figure 4 and 5 are improved.

Line 46. Typo, the family is "Pneumoviridae". The former subfamily was "Pneumovirinae". Table 3 (former Table 2) is still unclear. Didn't all patients eventually either improve and were discharged, or they died? So, what are the definitions of "severe", "not severe", or "high for improvement"? If these who were eventually discharged were categorized into severe and not severe, that is reasonable, but the authors should explain what criteria. And how does "not severe" differ from "high for improvement"? Please clarify in the text or table.

Author Response

The authors have responded nicely to most of my prior concerns. The clarification of the surveillance system is very helpful, as is the data on age distribution. The new figure 4 and 5 are improved.

Line 46. Typo, the family is "Pneumoviridae". The former subfamily was "Pneumovirinae".

Change Pneumovirinae by Pneumoviridae

Table 3 (former Table 2) is still unclear. Didn't all patients eventually either improve and were discharged, or they died? So, what are the definitions of "severe", "not severe", or "high for improvement"? If these who were eventually discharged were categorized into severe and not severe, that is reasonable, but the authors should explain what criteria. And how does "not severe" differ from "high for improvement"? Please clarify in the text or table.

We add Line 341-345. Table 3 shows the classification of hospitalized patients at the time of the sample process, it is important to mention that the status of hospitalized patients with severe acute respiratory infection is not always updated by clinicians, so it is unknown if non-serious patients (not intubated with severe acute respiratory infection) evolved to severe (intubated with pneumonia) or if severe patients left the hospital for improvement or died.
